# Intoxication of antibiotic persisters by host RNS inactivates their efflux machinery during infection

**Séverin Ronneau**\*, **Charlotte Michaux, Rachel T. Giorgio, Sophie Helaine**◉\*

Department of Microbiology, Harvard Medical School, Boston, Massachusetts, United States of America

\* severin_ronneau@hms.harvard.edu (SR); sophie_helaine@hms.harvard.edu (SH)

**Data Availability Statement:** All data are in the manuscript and/or supporting information files.

**Funding:** The author(s) received no specific funding for this work.

## Abstract

The host environment is of critical importance for antibiotic efficacy. By impacting bacterial machineries, stresses encountered by pathogens during infection promote the formation of phenotypic variants that are transiently insensitive to the action of antibiotics. It is assumed that these recalcitrant bacteria—termed persisters—contribute to antibiotic treatment failure and relapsing infections. Recently, we demonstrated that host reactive nitrogen species (RNS) transiently protect persisters against the action of β-lactam antibiotics by delaying their regrowth within host cells. Here, we discovered that RNS intoxication of persisters also collaterally sensitizing them to fluoroquinolones during infection, explaining the higher efficiency of fluoroquinolones against intramacrophage *Salmonella*. By reducing bacterial respiration and the proton-motive force, RNS inactivate the AcrAB efflux machinery of persisters, facilitating the accumulation of fluoroquinolones intracellularly. Our work shows that target inactivity is not the sole reason for *Salmonella* persisters to withstand antibiotics during infection, with active efflux being a major contributor to survival. Thus, understanding how the host environment impacts persister physiology is critical to optimize antibiotics efficacy during infection.

## Author summary

By influencing the physiology of bacterial pathogens, the host environment can either limit or potentiate the effectiveness of antibiotics during infection. Recently, we demonstrated that host reactive nitrogen species (RNS), generated by macrophages in response to *Salmonella* infection, can transiently shield a subset of recalcitrant cells from the effects of β-lactam antibiotics by reducing their cellular respiration. Here, we showed that although bacteria intoxicated by RNS are protected from β-lactam antibiotics, they remain highly susceptible to fluoroquinolones, another class of antibiotics. We found that by reducing cellular respiration, host RNS collaterally inactivate bacterial efflux machinery, which is an essential determinant of fluoroquinolone recalcitrance. Our study explains how the modulation of bacterial respiration by the host environment can differentially impact antibiotic effectiveness during infection. Understanding how host factors influence the physiology of pathogens is essential for optimizing antibiotic use.

**Competing interests:** The authors have no competing interests.

## Introduction

In contrast to resistant bacteria, antibiotic-tolerant cells do not grow in the presence of antibiotics and escape the action of bactericidal drugs without requiring any genetic changes [1,2]. Recent work has drawn attention to the contribution of this phenomenon in the recurrence of many bacterial infections, from life-threatening tuberculosis to common urinary tract infections [3–5]. By compromising the efficacy of the antibiotic treatment, these recalcitrant bacteria may lead to recurrent infections but also contribute to the selection and spreading of antibiotic resistance [6–9]. Therefore, understanding how these cells survive antibiotic exposure within their host should provide new insights to improve current treatments.

Although it is established that the host environment and the stresses it imposes on pathogens drive antibiotic recalcitrance, much is still to be learned about how it impacts antimicrobial efficacy during infection. Whereas in laboratory media, many conditions and mutations limit the action of bactericidal drugs by inactivating antibiotic targets, the physiological state adopted by bacteria surviving antibiotics during infection can be extremely different [10]. For example, bacterial populations treated with chloramphenicol, a translation inhibitor, enter a non-growing state which protects them against the action of β-lactams [11]. As opposed to these inactive cells equipped to survive laboratory conditions, recalcitrant bacteria formed in host cells often retain translational activity despite their growth arrest [12–17], indicating that active processes support antibiotic persistence during infection.

Previously, we have shown that internalization of *Salmonella enterica* serovar Typhimurium (henceforward referred to as *Salmonella*) by macrophages promotes the formation of phenotypic variants termed antibiotic persisters [18]. Those persisters are a subpopulation of non-growing bacteria that withstand antibiotic exposure and have the potential to repopulate their environment with antibiotic-sensitive cells when the treatment is ceased [19]. Recently, we demonstrated that following persister formation, host reactive nitrogen species (RNS) maintain intracellular *Salmonella* persisters in their non-growing state for an extended period of time by lowering bacterial respiration through intoxication of the tricarboxylic acid (TCA) cycle. By delaying the growth resumption of persisters, RNS transiently protect *Salmonella* from the action of β-lactams, which are inefficient at killing non-growing bacteria [19]. Intriguingly, a recent study reported that RNS-exposed *Salmonella* have a lower survival to fluoroquinolones in comparison with nonexposed cells during mouse infection [20]. This apparent contradiction led us to explore the impact of RNS on the susceptibility of *Salmonella* to fluoroquinolones. In our study, we find that whereas host RNS impede the action of β-lactam antibiotics, they synergize with fluoroquinolones. By limiting *Salmonella* cellular respiration, RNS adversely impact numerous cellular functions in bacteria, in particular, the generation and maintenance of the proton motive force (PMF) across the cell membrane. Such PMF is required to fuel many components of the bacterial efflux machineries, which have been associated with antibiotic recalcitrance [21–24]. We therefore assessed the downstream consequences of RNS intoxication on persister efflux activity during infection and demonstrate that RNS inactivate PMF-dependent efflux pumps, allowing fluoroquinolone accumulation in non-growing bacteria and greater antibiotic efficacy. Our findings highlight the complex role of the host environment on persister physiology during infection and its implication in antibiotic recalcitrance.

## Results

### Persisters intoxicated by host RNS are sensitive to ciprofloxacin

Internalization of *Salmonella* by immune cells promotes phenotypic heterogeneity within the intracellular population [18,25]. While most bacteria are actively proliferating, a subpopulation of cells remains in a non-growing state for extended periods of time [18,19]. Since β-lactam antibiotics are only effective against growing cells, intramacrophage *Salmonella* survive antibiotic treatment as long as they remain non-growing, explaining why persisters are exclusively found in the non-growing fraction of the population [11,19,26]. In agreement with our previous fluorescence dilution (FD) experiments in which bacteria are preloaded with an inducible GFP prior the infection, only bacteria maintaining their initial level of fluorescence (*i.e.* non-growers) survived 24 hours of treatment with cefotaxime, a β-lactam antibiotic (Fig 1A) [18,19]. We previously reported that these non-growing cells were also able to withstand other classes of antibiotics such as fluoroquinolones which retain some bactericidal activity against non-growing cells *in vitro* [18,27]. Here, we compared the efficacy of cefotaxime (β-lactam) and ciprofloxacin (fluoroquinolone) on intramacrophage *Salmonella* during infection. We infected bone-marrow-derived macrophages with wild-type (WT) *Salmonella* and treated them with antibiotics for 24 h (experimental approach is depicted on S1A Fig). In agreement with previous observations [18], we observed that ciprofloxacin exhibited greater killing efficacy than cefotaxime (Fig 1B), suggesting that a portion of the non-growing bacteria that survive β-lactam exposure remained susceptible to ciprofloxacin. In support of this hypothesis, the addition of cefotaxime to the ciprofloxacin treatment failed to enhance bacterial clearance in comparison with ciprofloxacin alone (Fig 1B).

Recently, we showed that host reactive nitrogen species (RNS) influence persister levels within host cells during β-lactam treatment by extending the time bacteria are maintained in a non-growing state during infection [19]. In support of this, stimulation of murine macrophages with interferon-gamma (IFN-ϒ) prior to the infection increases RNS production and the proportion of non-growing *Salmonella*. Conversely, the absence of RNS in $Nos2^{-/-}$ mice decreases the number of non-growers within macrophages [19]. Consistent with β-lactams ineffectiveness against non-growing cells [11,26], the number of persisters surviving cefotaxime (Fig 1C) correlates with RNS levels (S2 Fig) inside macrophages (experimental approach is depicted on S1B Fig) [19].

To assess if host RNS also impact *Salmonella* recalcitrance to fluoroquinolones, we compared the efficacy of 24 h cefotaxime and ciprofloxacin treatment on infected macrophages producing different levels of RNS (S1B and S2 Figs). In agreement with our previous study, we found that the number of persisters surviving cefotaxime treatment was the highest in IFN-ϒ-stimulated macrophages and the lowest in RNS-deficient $Nos2^{-/-}$ macrophages (Figs 1C and S2) [19]. The absence of complete clearance observed in $Nos2^{-/-}$ macrophages during cefotaxime treatment indicates that some RNS-independent persisters exist within host cells. In contrast with cefotaxime treatment, no noticeable difference between conditions was observed after ciprofloxacin exposure (Fig 1C). Moreover, the number of persisters recovered after ciprofloxacin and cefotaxime was similar in the absence of RNS production in $Nos2^{-/-}$ macrophages (Fig 1C), suggesting that RNS-dependent persisters present in WT macrophages are sensitive to ciprofloxacin.

To confirm that ciprofloxacin eradicated RNS-dependent persisters, we sequentially treated infected macrophages with cefotaxime and then ciprofloxacin. We first infected macrophages, IFN-ϒ-stimulated or not, with WT *Salmonella* and treated with cefotaxime to select for persisters. After 24 h, we replaced cefotaxime with ciprofloxacin and monitored the amount of bacteria recovered after 24 additional hours (experimental approach is depicted on S1C Fig).

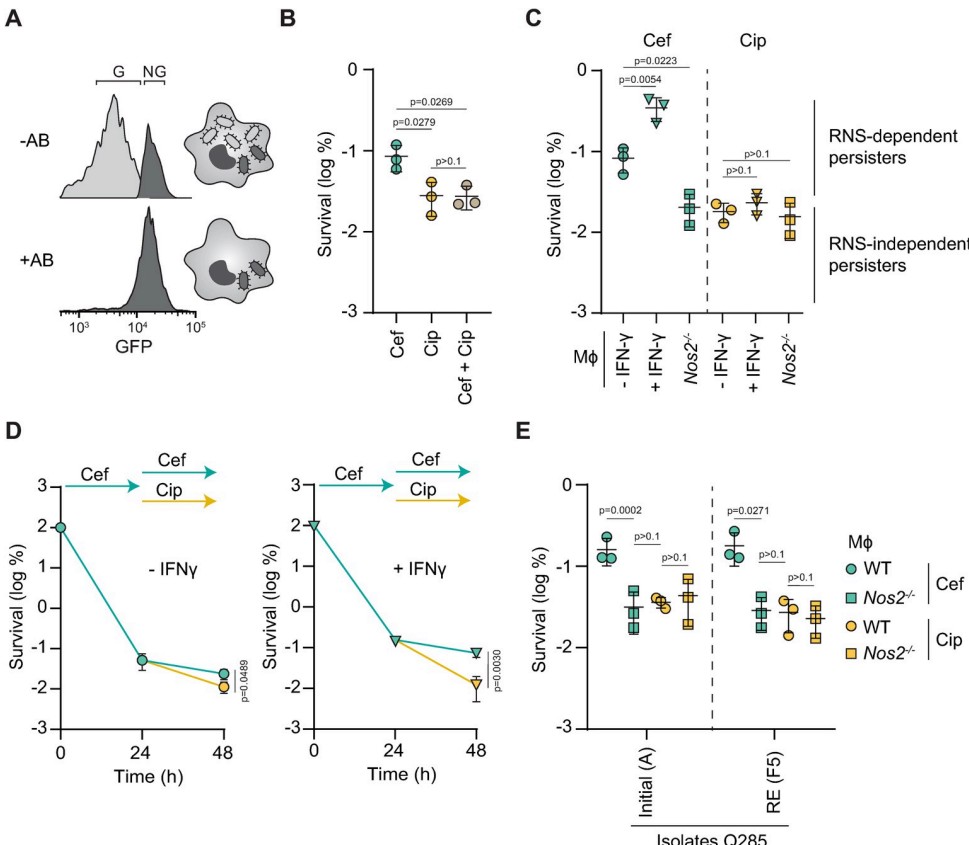

**Fig 1. RNS-dependent persisters are sensitive to ciprofloxacin.** (**A**) Representative fluorescence-activated cell sorting (FACS) plots of fluorescence dilution experiments tracking the proportion of growing (G; light gray) and non-growing (NG; dark gray) WT *Salmonella* recovered from WT macrophages (Mφ) in the presence (+AB) or in the absence (-AB) of cefotaxime at 16 h of infection. (**B**) 24 h cefotaxime and/or ciprofloxacin survival of WT *Salmonella* in WT Mφ normalized to values after 30 min internalization. *p* values are indicated (ANOVA with Tukey's correction for multiple comparisons); error bars depict means and standard deviation (SD); n = 3. (**C**) 24 h cefotaxime (light blue) or ciprofloxacin (yellow) survival of WT *Salmonella* in unstimulated (-IFN-γ) or IFN-γ-stimulated (+IFN-γ) WT or *Nos2*[-/-] Mφ normalized to values after 30 min internalization. Distinction between RNS-dependent and independent persisters was determined using the proportion of persisters in *Nos2*[-/-] Mφ. *p* values are indicated (ANOVA with Dunnett's correction for multiple testing against the–IFN-γ condition); error bars depict means and standard deviation (SD); n = 3. (**D**) Persister clearance after 24 h of cefotaxime or ciprofloxacin treatment following 24 h of cefotaxime treatment of WT *Salmonella* in unstimulated (-IFN-γ) or IFN-γ-stimulated (+IFN-γ) WT Mφ normalized to values after 30 min internalization. *p* values are indicated (unpaired t test at 48h); error bars depict means and standard deviation (SD). (**E**) 24 h cefotaxime (light blue) or ciprofloxacin (yellow) survival of the initial (Q285A) and recurrent (Q285F5) isolates of ST313 *Salmonella* Q285 in WT or *Nos2*[-/-] Mφ normalized to values after 30 min internalization. *p* values are indicated (ANOVA with Tukey's correction for multiple comparisons); error bars depict means and standard deviation (SD); n = 3.

Consistent with our hypothesis, ciprofloxacin was more effective than cefotaxime, especially inside IFN-ϒ-stimulated macrophages where most persisters are RNS-dependent (Fig 1C–1D) [19]. These data suggest that RNS-dependent persisters only survive cefotaxime whereas RNS-independent persisters survive both cefotaxime and ciprofloxacin.

To determine if host RNS also impact antibiotics efficacy against clinically-relevant strains, we tested two invasive non-typhoidal *Salmonella* (iNTS) clinical isolates recovered from a HIV-positive patient experiencing a recurrent infection: Q285A (isolated from the first episode of disease) and Q285F5 (a subsequent derivative isolated after several rounds of antibiotic treatment) [28]. As shown previously for the laboratory strains, the absence of RNS in *Nos2*[-/-]

macrophages increased cefotaxime efficacy against both isolates, supporting a role of host RNS in β-lactam recalcitrance in clinically-relevant strains (Fig 1E). More importantly, treatment with ciprofloxacin resulted in killing similar to cefotaxime in $Nos2^{-/-}$ macrophages (Fig 1E), supporting that the sensitivity of RNS-intoxicated persisters to fluoroquinolones also applies to clinically-relevant strains.

## The AcrAB-TolC efflux machinery supports *Salmonella* ciprofloxacin persistence in macrophages

Our results suggested that RNS-intoxicated persisters that survive cefotaxime are unable to perform an essential function required to withstand ciprofloxacin. Previous studies have shown that in laboratory medium, lowering the intracellular concentration of antibiotics through the bacterial efflux systems contributes to persister survival [21,29]. We thus wondered if efflux machineries also contribute to *Salmonella* antibiotic persistence within macrophages. To explore this possibility, we evaluated the impact of the loss of TolC, a critical outer membrane channel required for the efflux of many toxic compounds, including fluoroquinolones [30]. We compared the ability of the WT and a *tolC* deletion mutant to survive 48 hours of antibiotic exposure in macrophages. We found that loss of *tolC* drastically reduced the number of persisters after treatment with ciprofloxacin but not cefotaxime, highlighting the critical role of the efflux machineries in the ability of persisters to survive fluoroquinolones (Fig 2A). Since TolC interacts with a large array of inner membrane transporters (Fig 2B) [30], we tested the ability of single knock-out mutants of functional partners of TolC (AcrB, AcrD, AcrF, MdsB, MdtC, and EmrB) to survive 48 hours ciprofloxacin exposure within macrophages. We found that the *acrB* mutant displayed a lower persister fraction than the WT strain, similar to that of the *tolC* mutant after ciprofloxacin exposure (Fig 2C). To exclude the possibility that the *acrB* deletion indirectly affects persister levels by impeding general bacterial virulence during infection, we compared the proliferation of WT and *ΔacrB* strains in macrophages in absence of antibiotics. We found that the overall intracellular proliferation after 16 hours of infection was similar in the WT and the *acrB* mutant (Fig 2D), suggesting that AcrB directly contributes to antibiotic recalcitrance rather than more generally bacterial behavior within host cells.

Using fluorescence accumulation [31], we previously showed that intramacrophage persisters retain transcriptional and translational activity, which maximizes their survival inside the host [12,18,19]. To investigate the role of AcrB in persister physiology, we first exposed macrophages infected with WT or the *acrB* mutant to 24 hours of cefotaxime to select for non-growing bacteria [11,18,19]. Then, we exposed the macrophages to cefotaxime or ciprofloxacin for an additional 24-hour period and assessed the proportion of active and inactive non-growing bacteria in the population by inducing the production of GFP, which can only be synthesized by the active subpopulation (Fig 2E). Consistent with our hypothesis that *acrB* contributes to persister survival during ciprofloxacin treatment, the loss of *acrB* strongly reduced the amount of active non-growers (aNG) and delayed GFP accumulation in the population exposed to ciprofloxacin but not cefotaxime (Fig 2E). These results reveal the critical role of the AcrAB-TolC machinery on persister survival during ciprofloxacin exposure.

## Host RNS sensitize persisters to ciprofloxacin by reducing their efflux activity during infection

Since the activity of the AcrAB-TolC tripartite module is fuelled by the PMF [32,33], we postulated that by impacting the TCA cycle and cellular respiration, host RNS could collaterally inactivate the efflux activity of RNS-dependent persisters, sensitizing them to fluoroquinolones

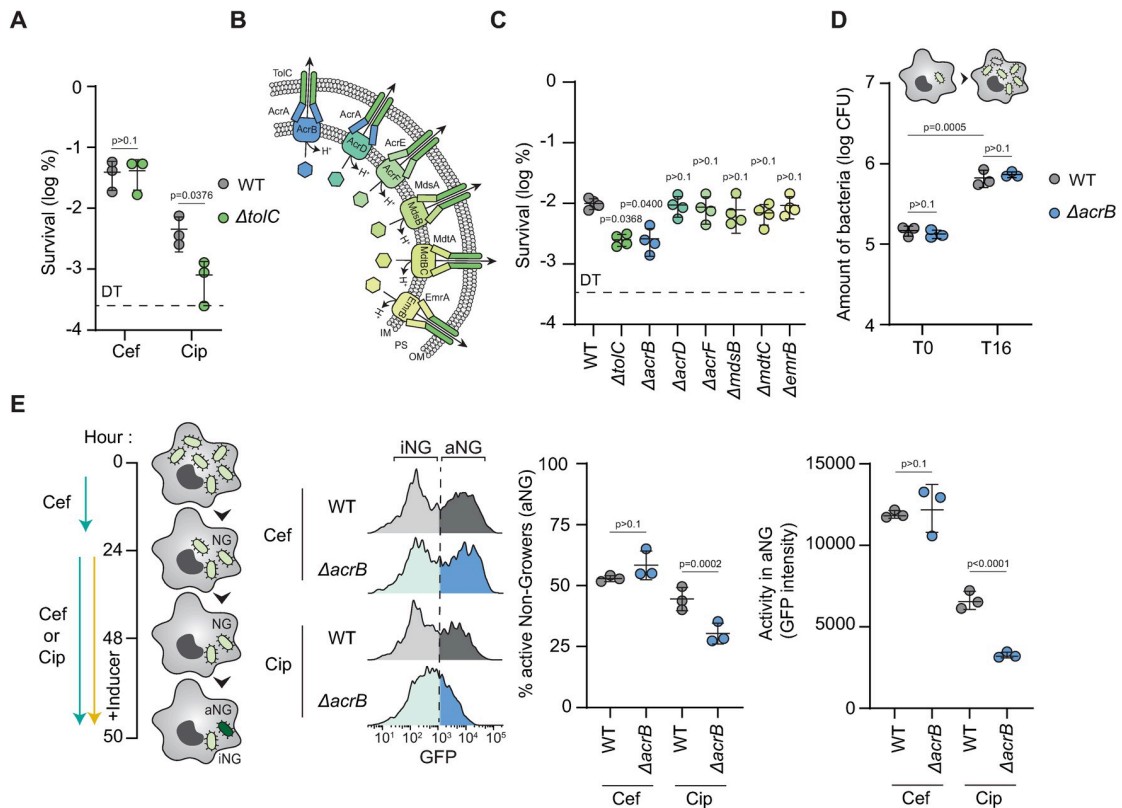

**Fig 2. AcrAB-TolC efflux machinery contributes to persister survival during ciprofloxacin treatment within host cells.** (**A**) 48 h cefotaxime or ciprofloxacin survival of WT (gray) or Δ*tolC* (green) *Salmonella* in WT Mφ normalized to values after 30 min internalization. *p* values are indicated (ANOVA with Tukey's correction for multiple comparisons); error bars depict means and standard deviation (SD); n = 3. DT: Detection Threshold. (**B**) Illustration of TolC-dependent efflux machineries of *Salmonella*. IM: Inner Membrane, PS: Periplasmic Space, OM: Outer Membrane. (**C**) 48 h ciprofloxacin survival of WT, Δ*tolC*, Δ*acrB*, Δ*acrD*, Δ*acrF*, Δ*mdsB*, Δ*mdtC* or Δ*emrB Salmonella* in WT Mφ normalized to values after 30 min internalization. *p* values are indicated (ANOVA with Dunnett's correction for multiple testing against the WT); error bars depict means and standard deviation (SD); n = 4. DT: Detection Threshold. (**D**) Bacterial load of WT or Δ*acrB Salmonella* in WT Mφ after 30 min internalization (T0) and at 16 h of infection (T16) in the absence of antibiotics. *p* values are indicated (ANOVA with Tukey's correction for multiple comparisons); error bars depict means and standard deviation (SD); n = 3. (**E**) (Left) Illustration of the experimental setup. After 24 h of cefotaxime exposure, infected macrophages containing non-growers (NG) were exposed to 26 h of cefotaxime or ciprofloxacin. To distinguish active (aNG) and inactive (iNG) non-growers, production of GFP was induced during 2 h prior extraction and analysis. (Right) Representative FACS plots and quantification of the level of transcriptional/translational activity in active and inactive cefotaxime or ciprofloxacine-treated intramacrophage WT or Δ*acrB Salmonella* in WT Mφ at 50 h of infection. *p* values are indicated (ANOVA with Tukey's correction for multiple comparisons); error bars depict means and standard deviation (SD); n = 3.

(Fig 3A). To assess whether the ciprofloxacin susceptibility of RNS-intoxicated bacteria was caused by the impact of host RNS on bacterial respiration, we used a *sucB* knock-out, which abolishes the activity of the α-ketoglutarate dehydrogenase (αKDH) enzymatic complex of the TCA cycle [19]. The lack of a functional TCA cycle in a *sucB* mutant increases the fraction of non-growing bacteria with low cellular respiration, mimicking an extreme intoxication of the WT strain by host RNS [19]. Consequently, the number of bacteria surviving cefotaxime exposure is similar between a *sucB* mutant in unstimulated macrophages and the WT strain exposed to high level of RNS in IFN-ϒ-stimulated macrophages (Figs 1C and 3B) [19]. Similar to WT (Fig 1B), most *sucB* mutant persisters were eradicated by the action of ciprofloxacin (Fig 1C), suggesting that persisters with a low cellular respiration are sensitive to ciprofloxacin. To confirm our hypothesis, we sequentially treated macrophages infected with the *sucB*

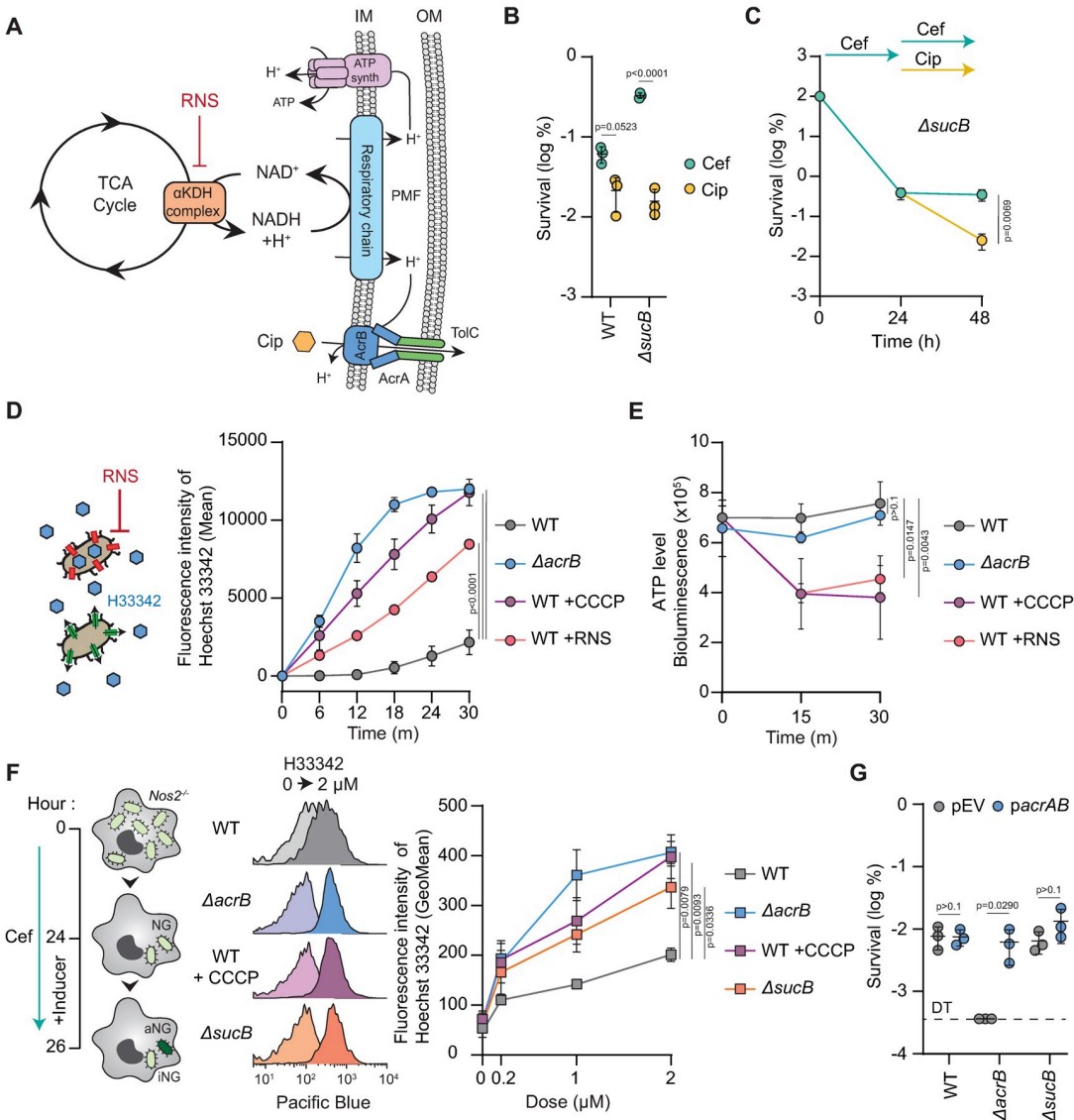

**Fig 3. Host RNS inactivates efflux activity of intramacrophage persisters.** (**A**) Model of intoxication of the TCA cycle by host RNS. Corruption of the α-KDH complex limits cellular respiration of persisters, which limit the proton-motive force (PMF). Consequently, host RNS limits ATP production and efflux activity of intramacrophage persisters. (**B**) 24 h cefotaxime (light blue) or ciprofloxacin (yellow) survival of WT or *ΔsucB Salmonella* in WT Mφ normalized to values after 30 min internalization. *p* values are indicated (ANOVA with Tukey's correction for multiple comparisons); error bars depict means and standard deviation (SD); n = 3. (**C**) Persister clearance after 24 h of cefotaxime followed by 24 h of cefotaxime or ciprofloxacin treatment of *ΔsucB Salmonella* in WT Mφ normalized to values after 30 min internalization. *p* values are indicated (unpaired t test at 48h); error bars depict means and standard deviation (SD). (**D**) (Left) Illustration of the experimental setup. Efflux-dependent accumulation of Hoechst (H33342) was assessed in absence or in presence of RNS. (Right) Measurement of H33342 accumulation over time in WT or *ΔacrB Salmonella* in presence or in absence of CCCP or RNS. *p* values are indicated (ANOVA with Dunnett's correction for multiple testing against the WT at 30m); error bars depict means and standard deviation (SD). (**E**) ATP level over time obtained by bioluminescence. Experimental conditions are the same as in panel **D**. *p* values are indicated (ANOVA with Dunnett's correction for multiple testing against the WT at 30m); error bars depict means and standard deviation (SD). (**F**) (Left) Illustration of the experimental setup. After 24 h of cefotaxime exposure, production of GFP was induced in intramacrophage non-growers for 2 h to distinguish active (aNG) and inactive (iNG) non-growers. Then, the efflux activity of aNG was assessed using H33342 dye. (Right) Representative FACS plots and quantification of the efflux activity of WT, *ΔacrB* or *ΔsucB Salmonella* in *Nos2*[-/-] Mφ. As a control, WT persisters were also tested in the presence of CCCP, an inhibitor of cellular respiration. *p* values are indicated (ANOVA with Dunnett's correction for multiple testing against the WT treated with 2 μM H33342 dye); error bars depict means and standard deviation (SD). (**G**) 48 h ciprofloxacin survival of WT, *ΔacrB* or *ΔsucB Salmonella* in WT Mφ normalized to values after 30 min internalization. WT, *ΔacrB* and *ΔsucB* strains were complemented with an empty vector (pEV) or *acrAB* (p*acrAB*). *p* values are indicated (ANOVA with Tukey's correction for multiple comparisons); error bars depict means and standard deviation (SD); n = 3. DT: Detection Threshold.

mutant with cefotaxime and then ciprofloxacin (experimental approach is depicted on S1C Fig). In agreement with what we observed with IFN-ϒ-stimulated macrophages for the WT strain (Fig 1B), most *sucB* cefotaxime persisters were eliminated by ciprofloxacin (Fig 3C). Collectively, these data suggest that by reducing cellular respiration, host RNS inactivate efflux machineries of intramacrophage persisters.

We tested this hypothesis first in M9 minimal medium containing citrate as the sole carbon source. We previously showed that growth on citrate required a functional TCA cycle that can be inhibited by the addition of exogenous RNS [19]. Taking advantage of Hoechst 33342 (H33342), an AcrAB substrate that becomes fluorescent when bound to dsDNA [34,35], we quantified the effect of Spermine NONOate, a nitric oxide-releasing compound, on the accumulation of the H33342 dye in *Salmonella*. As expected, lack of a functional AcrAB-TolC complex led to the rapid accumulation of H33342 in Δ*acrB* compared to the WT strain (Figs 3D and S3A). Similarly, addition of NONOate to the WT strain significantly impaired its efflux activity (Fig 3D). As a control, we used CCCP, a proton-ionophore that dissipates the PMF and thereby impacts efflux activities [36,37]. CCCP nearly abolished H33342 efflux by WT *Salmonella* (Fig 3D). Both CCCP and RNS also reduced *Salmonella* ATP levels in these conditions, supporting a role for RNS in interfering with efflux activity through lowering the PMF (Fig 3E).

To further explore the relationship between cellular respiration and efflux activity of persisters during infection, we extracted active non-growers of the WT, Δ*sucB*, and Δ*acrB* strains after 26 hours of cefotaxime treatment inside *Nos2*[-/-] macrophages. Then, active non-growers were exposed to different concentrations of H33342 (0, 0.2, 1, or 2 μM) to evaluate their efflux activity (Fig 3F). As observed at the population-level *in vitro* (Fig 3D), persisters devoid of *acrB* were unable to efflux H33342 in contrast with the RNS-independent persisters of the WT strain extracted from infected *Nos2*[-/-] macrophages (Fig 3F). When cellular respiration of WT persisters was lowered with CCCP, their efflux activity was drastically reduced (Fig 3D). Similarly, the *sucB* mutant, which mimics a constitutive RNS intoxication in *Nos2*[-/-] macrophages [19], displayed a reduced efflux activity (Fig 3F). In agreement with these results, intramacrophage persisters of IFN-ϒ-stimulated macrophages also exhibited a lower efflux activity than persisters released from *Nos2*[-/-] macrophages (S3B Fig).

Our results suggest that the sensitivity of RNS-intoxicated persisters to ciprofloxacin is caused by a reduced efflux activity due to a lower PMF. To validate our hypothesis, we overexpressed *acrAB* in the WT, *acrB*, and *sucB* mutants and tested their survival to 48 hours of ciprofloxacin exposure inside macrophages. If host RNS inactivate the efflux machineries of persisters, overexpression of AcrAB should not be beneficial in the WT or in the *sucB* mutant. Although the overexpression of *acrAB* can complement the defect of an *acrB* mutant, it did not increase the survival of the WT or the *sucB* mutant to ciprofloxacin (Fig 3G). Collectively, these data support that host RNS collaterally inactivate the efflux activity of persisters, rendering them vulnerable to ciprofloxacin during infection.

## Discussion

The host environment is a critical determinant of antibiotic efficacy [2,38,39]. However, little is known about which host factors synergize or antagonize the action of antimicrobials during infection. Previously, we showed that internalization of *Salmonella* by macrophages promotes the formation of antibiotic persisters [18]. More recently, we demonstrated that host RNS protect persisters from the action of β-lactams by lowering the cellular respiration of persisters through TCA cycle intoxication, maintaining them in a non-growing state for an extended

period of time [19]. Here, we show that host RNS can collaterally sensitize *Salmonella* persisters to fluoroquinolones by corrupting their PMF-dependent efflux machinery (Fig 4).

The ability of persisters to withstand antibiotics has often been attributed to the inactivity of antibiotic targets in a subpopulation of cells [40]. For example, β-lactams, which target the cell wall machinery, are only effective against actively growing cells [11,41]. In contrast, fluoroquinolone persistence cannot be explained by growth arrest or reduced target activity alone and involves many other factors, explaining why these antibiotics are unable to fully eradicate non-growing bacteria *in vitro* [42]. Since this class of antibiotics mediates their bactericidal activity through the accumulation of DNA damage, non-growing bacteria harboring multiple copies of the bacterial chromosome and displaying active DNA-repair are more likely to survive fluoroquinolone exposure [42–44]. Moreover, efficient efflux activity also mitigates the impact of DNA-targeting agents on genome integrity by limiting the accumulation of antimicrobial drugs inside bacteria [21,29,45]. In agreement with this, our work emphasizes the critical role of the AcrAB-TolC tripartite complex in ciprofloxacin persistence of *Salmonella* during macrophage infection (Fig 2). In the absence of efflux activity, we demonstrate that ciprofloxacin effectively eliminates most intramacrophage persisters, which aligns with our previous observations indicating that persisters actively accumulate DNA during infection [44].

It was previously shown that *in vitro* persisters show a significantly higher expression of a large number of multi-drug efflux-associated genes than their drug-sensitive counterparts [21]. Although we do not exclude that internalization of *Salmonella* inside macrophages or the presence of antibiotics itself might enhance the expression of efflux pumps-encoding genes, overexpression of *acrAB* alone fails to protect intramacrophage persisters from the action of ciprofloxacin (Fig 3F). This suggests that even if non-growing bacteria enriched for efflux machineries are more likely to survive to antibiotic exposure, the impact of the host environment on their pumping activity is a critical determinant in their survival. In support of this, we found that RNS-dependent persisters that display a lower cellular respiration due to the corruption of their TCA cycle are unable to efficiently fuel their efflux machinery, ultimately leading to their eradication during ciprofloxacin treatment (Fig 4B).

Intriguingly, we found some persisters still survive ciprofloxacin exposure in the presence of high levels of RNS in IFN-Υ-stimulated macrophages (Fig 1C–1D). Since most of these persisters still require a functional AcrAB to withstand fluoroquinolone treatment (Fig 2C), it suggests that they can maintain the activity of their efflux pumps, irrespectively of the presence of a functional TCA cycle. One simple explanation would be that these cells rely on alternative metabolic pathways to fuel their PMF, bypassing the intoxication of their TCA cycle by host RNS. This is supported by the presence of persisters surviving ciprofloxacin treatment in the *sucB* mutant, which is devoid of a functional TCA cycle (Fig 3B–3C) [19]. Although nutrient availability is usually scarce during infection, host cells offer an important diversity of substrates and parallel exploitation of host nutrients may allow *Salmonella* to be extremely resilient against metabolic perturbations [46–48]. The maintenance of a basal metabolism in non-growing *Salmonella* is not only required to maintain efficient efflux activity but also to power energy-consuming activities such as transcription, translation (Fig 2E), DNA repair [44], and secretion of type three secretion system effectors [12]. Accordingly, the utilization of efflux pump inhibitors such as the phenylalanine-arginine β-naphthylamide (PAβN) potentiates the activity of antimicrobials on recalcitrant cells [21,23]. Uncovering metabolic pathways that support cellular respiration of persisters inside host cells might allow the identification of new adjuvants that foster the activity of fluoroquinolones on persisters by limiting their PMF during infection.

Altogether, our work shows that, while the efflux machinery is a primary determinant of fluoroquinolone efficacy, its activity is strongly influenced by a combination of host factors,

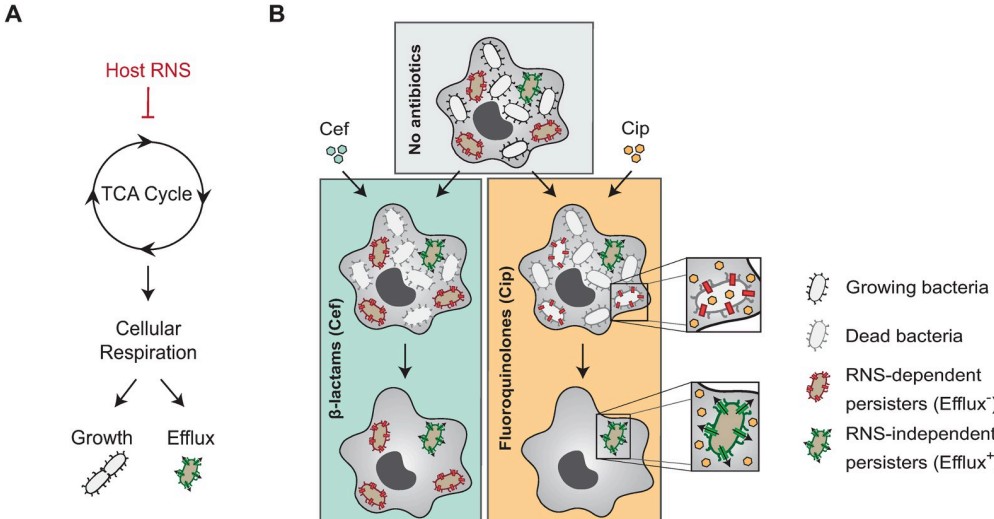

**Fig 4. Model.** (**A**) Host RNS intoxicate the TCA cycle of persisters, thereby limiting their cellular respiration while residing within host cells. Consequently, intramacrophage persisters intoxicated with RNS remain in a growth-arrested state but are unable to sustain efficient efflux activity. (**B**) Intramacrophage population of *Salmonella* displays strong phenotypic heterogeneity. In absence of antibiotics, most bacteria are actively growing (gray) whereas a subpopulation of cells remains in a non-growing state (brown). Treatment with cefotaxime (cef) results in the killing of the growing population but is ineffective against non-growers. In contrast, ciprofloxacin eliminates not only the growing population but also non-growers that are unable to maintain an efficient efflux activity. Since RNS-dependent persisters cannot sustain such activity, they are unable to withstand ciprofloxacin exposure during infection.

such as RNS, and the metabolic status of each individual bacterium (Fig 4B). This multifactorial modulation of efflux activity results in heterogeneity in the pumping activity of intramacrophage *Salmonella*, explaining the incomplete clearance of the pathogen after antibiotic treatment (Fig 4B). Our model emphasizes that a better understanding of the impact of the host environment on bacterial respiration sheds light on persister vulnerabilities during infection and may ultimately lead to the development of better strategies for the treatment of recalcitrant infections.

## Materials and methods

### Ethics statement

The work carried out here has been approved by the Harvard Committee on Microbiological Safety. All experiments involving mice were pre-reviewed and approved by the Harvard Medical School Institutional Animal Care and Use Committee (IACUC).

### Bacterial strains and plasmids

Oligonucleotides, strains and plasmids used in this study are listed in S1, S2 and S3 Tables. *Escherichia coli* DH5α and *Salmonella* strains used in this study were grown aerobically at 37˚C in Luria-Bertani (LB) broth (Invitrogen) or in M9 minimal medium (1x M9 salts, Sigma; 2 mM $MgSO_4$, Sigma; 0.1 mM $CaCl_2$, Sigma) supplemented with 0.1% casamino acids (BD), 0.2% glucose (Sigma), 0.2% glycerol (Sigma), or 0.2% citrate (VWR) when indicated. Antibiotics were used at the following concentrations: carbenicillin (100 μg/ml, CHEM-IMPEX INT'L INC), kanamycin (50 μg/ml, Sigma), and chloramphenicol (34 μg/ml, CHEM-IMPEX INT'L INC). For previously published deletion mutants [49], the genotype of the deletion mutant was

transduced into wild-type *Salmonella enterica* serovar Typhimurium strain 12023/14028 (RS1) using P22 bacteriophage [50]. Successful transduction was then confirmed by PCR.

## Bone marrow-derived macrophages derivation and culture

Extraction and culture of bone marrow derived macrophages (BMDM) were performed as previously described [11,19]. Tibia and femurs of C57BL/6 (WT) or B6.129P2-Nos2<tm1Lau>/J (*Nos2*<sup>-/-</sup>) female mice (Jackson Lab) older than 8 weeks were first isolated from both legs. Bone heads were cut and bone marrow was flushed out using a 23Gx3/4 needle (BD). Red blood cells were lysed in freshly prepared 0.83% $NH_4Cl$ (Sigma) for 3 min. The remaining cells were seeded in 100 mm non-tissue culture treated plates (Corning) at a concentration of 3E+6 cells per plate in 8 ml of differentiation medium based on Dulbecco's modified eagle medium with high glucose (DMEM; Corning) supplemented with 20% (vol/vol) of L929 supernatant (LCM), 10% (vol/vol) of fetal bovine serum (FBS; Premium Select from R&D Systems), 10 mM of HEPES (Sigma), 1 mM of sodium pyruvate (Sigma), 0.05 mM of β-mercaptoethanol (Sigma) and 100 U/ml of penicillin/streptomycin (Genesee Scientific). After 3 days, 10 ml of fresh differentiation medium was added to each plates. On day 7, the fully differentiated BMDMs were collected. Macrophages were then seeded 24 hours prior infection in infection medium (DMEM medium supplemented with 10% FBS, 10 mM of HEPES, 1mM of sodium pyruvate and 0.05 mM β-mercaptoethanol) in 6-well tissue culture treated plates at a concentration of either 1E+6 macrophages per well if freshly harvested or 1.2E+6 macrophages per well if from frozen stock. When indicated, IFN-γ at the concentration of 50 ng/ml was added during the seeding process. Mice were housed with sterile bedding and nesting and received autoclaved chow and water over the course of the study.

## Macrophage infections and bacterial extraction

Macrophage infections and bacterial extraction were performed as previously described (*3*). Briefly, bacteria were grown in LB or M9 minimal medium for 16 hours. Stationary phase bacteria were opsonized for 20 min (170 μl of infection medium, 20 μl of mouse serum (Sigma), and 45 μl of bacteria) and added to the macrophages at a Multiplicity of Infection (MOI) of 10. Synchronization of the infection was performed by centrifugation (5 min; 110 x *g*; RT). To allow bacterial internalization, infected macrophages were then incubated at 37˚C with 5% $CO_2$. After 30 min, macrophage medium was exchanged with fresh medium containing either cefotaxime (100 μg/ml; TCI) to test intramacrophage bacterial antibiotic survival or gentamicin (50 μg/ml for the first 30 min then replaced and kept at 10 μg/ml; Sigma) to assess intramacrophage bacterial proliferation, respectively. At selected timepoints, infected macrophages were washed three times with PBS (Growcells) and lysed with a diluted solution of 0.1% Triton X-100 (Sigma) to extract intracellular bacteria. Bacteria were collected, centrifuged for 3 min at 16,000 x *g* and resuspended in PBS prior further experiments.

## Fluorescence dilution analysis of intramacrophage *Salmonella*

Fluorescence dilution experiments presented in Fig 1A were performed as previously described [19]. Briefly, a WT *Salmonella* strain harbouring the plasmidic fluorescence dilution reporter (pFCcGi) was grown overnight in M9 minimal medium with carbenicillin and 0.4% arabinose as sole carbon source resulting in the production of the green fluorescent protein (GFP) production. Bacteria preloaded with GFP were used to infect macrophages. After 30 min of internalization, macrophage medium was exchanged with fresh infection medium containing either cefotaxime (100 μg/ml) or gentamycin (50 μg/ml for the first 30 min then replaced and kept at 10 μg/ml to allow bacterial proliferation inside macrophages). In each

sample, constitutive mCherry was used to discriminate bacteria from debris. After 16 hours, bacteria were extracted from macrophages as previously described and stored at 4°C in PBS prior to flow cytometry analysis on a BD LSR II.

### *Salmonella* antibiotic survival and proliferation assays in infected macrophages

Antibiotic survival and proliferation assays were performed as previously described [11,19]. Briefly, part of the infected macrophages were used to assess bacterial invasion before antibiotic treatment as described above. Ten-fold serial dilutions in PBS were performed and drops of 20 µl were plated on LB agar to determine the number of CFU at T0. The rest of the infected macrophages received antibiotic treatment by medium exchange supplemented with cefotaxime (100 µg/ml), ciprofloxacin (5 µg/ml) or gentamycin (50 µg/ml for the first 30 min then replaced and kept at 10 µg/ml to allow bacterial proliferation inside macrophages). At indicated timepoints, bacteria from infected macrophages were extracted as described above and plated on LB agar to assess antibiotic survival. For the experiment presented on Fig 3G, expression of *acrAB* relied on the basal activity of the P*lac* promoter of the pCA24N plasmid. Complementation of the *sucB* mutant with the pCA24N_*sucB* vector was previously described in the same experimental conditions [19].

### Quantification of nitrite

Quantification of nitrite presented in S2 Fig was performed as described previously [19]. Briefly, WT or *Nos2*$^{-/-}$ macrophages used for nitrite quantification were infected with wild-type *Salmonella* and treated with cefotaxime. After 24 hours, nitrite quantifications were performed on supernatants using the Nitrite Assay kit (BioVision), following the manufacturers protocol. Nitrite standards were joined for each measurement replicates and based on the obtained values, a nitrite standard was used to calculate the nitrite concentration of each supernatant.

### Fluorescence accumulation of intramacrophage *Salmonella*

Fluorescence accumulation experiments presented in Fig 2E were performed as previously described [19]. Briefly, *Salmonella* strains (WT or *ΔacrB)* containing the pFCcGi reporter were grown overnight in LB with carbenicillin. After 24 hours of macrophage infection in presence of cefotaxime, 0.2% of arabinose was added to the infection medium. After 2 hours, bacteria were extracted from macrophages as previously described and stored at 4°C in PBS prior to flow cytometry analysis on a BD LSR II.

### Efflux activity assay *in vitro*

Bacteria were grown overnight in M9 minimal medium containing 0.2% glucose and 0.2% glycerol (M9GG). Then, 20 µl of the overnight culture were diluted in 180 µl of minimal medium containing 0.1% casamino acids, 0.2% citrate and 2 µM of Hoechst 33342 in a flat-bottom 96-well plate (Greiner). M9 minimal medium was supplemented with 1 mM of Spermine NONOate (Enzo Life Sciences) or 100 µM of carbonyl cyanide *m*-chlorophenylhydrazone (CCCP; Sigma) when indicated. Fluorescence was then monitored every 6 minutes for 30 minutes at 37°C by a plate reader (Tecan LifeScience).

## ATP measurement *in vitro*

Bacteria were grown overnight in M9 minimal medium containing 0.2% glucose and 0.2% glycerol (M9GG). 50 μl bacterial culture was diluted into 450 μl of minimal medium containing 0.1% casamino acids and 0.2% citrate. M9 minimal medium was supplemented with 1 mM of Spermine NONOate (Enzo Life Sciences) or 100 μM of carbonyl cyanide *m*-chlorophenylhydrazone (CCCP; Sigma) when indicated. Then, a sample was taken after 0, 30 or 60 min. The ATP level of each sample was measured using the BacTiter Glo Kit (Promega) according to the manufacturer's instructions.

## Efflux activity assay on intramacrophage persisters

*Salmonella* strains (WT, *ΔacrB*, or *ΔsucB)* containing the pFCcGi reporter were grown overnight in LB with carbenicillin. After 24 hours of macrophage infection in presence of cefotaxime, 0.2% of arabinose was added to the infection medium to induce the production of GFP. After 2 hours, bacteria were extracted from macrophages as previously described and resuspended in 1 ml of PBS containing 0, 0.2, 1 or 2 μM of Hoechst 33342. Bacteria were then cultivated for 30 minutes at 37˚C with shaking in a thermomixer (Eppendorf). When indicated, PBS was supplemented with 100 μM of carbonyl cyanide *m*-chlorophenylhydrazone (CCCP; Sigma). Finally, all samples were placed on ice and analysed by flow cytometry on a BD LSR II.

## Supporting information

**S1 Fig. Illustration of the experimental setups. (A)** Bone marrow-derived macrophages from WT mice were infected with WT *Salmonella*. Then, infected Mφ were treated with cefotaxime, ciprofloxacin or both for 24 h. Finally, persisters (in brown) were extracted and plated on LB agar plate for counting. (**B**) Bone marrow-derived macrophages from WT or *Nos2*[-/-] mice were cultivated in the absence or in the presence of IFN-γ and infected with WT *Salmonella*. Then, infected Mφ were treated with cefotaxime or ciprofloxacin for 24 h. Finally, persisters (in brown) were extracted and plated on LB agar plate for counting. (**C**) Infected bone marrow-derived macrophages from WT mice were treated for 24 h with cefotaxime to select persisters. Then, infected macrophages were treated with cefotaxime or ciprofloxacin for 24 additional hours. Finally, persisters (in brown) were extracted and plated on LB agar plate for counting. (TIF)

**S2 Fig. Production of RNS by host cells.** Quantification of nitric oxide production by macrophages was achieved by quantifying its stable byproduct nitrite in the infection medium. Nitrite concentration in the infection medium of unstimulated (circle) or IFN-γ-stimulated (triangle) WT and *Nos2*[-/-] (square) Mφ infected for 24 h with WT Salmonella and treated with cefotaxime. *p* values are indicated (ANOVA with Dunnett's correction for multiple testing against the—IFN- γ condition); error bars depict means and standard deviation (SD); n = 3. (TIF)

**S3 Fig. AcrAB-dependent efflux is limited by host RNS.** (**A**) Representative FACS plots and quantification of the efflux activity of WT or *ΔacrB* complemented with an empty vector (pEV) or *acrAB* (p*acrAB*) treated with 2 μM H33342 dye. *p* values are indicated (ANOVA with Tukey's correction for multiple comparisons); error bars depict means and standard deviation (SD); n = 3. Experimental conditions are the same as in Fig 3D. (**B**) Representative FACS plots and quantification of the efflux activity of WT *Salmonella* in *Nos2*[-/-] (gray) or IFN-γ-stimulated WT (red) Mφ. *p* value is indicated (unpaired t test); error bars depict means and standard deviation (SD). Experimental conditions are the same as in Fig 3F. (TIF)

**S1 Table. Primers used in this study.**
(XLSX)

**S2 Table. Plasmids used in this study.**
(XLSX)

**S3 Table. Strains used in this study.**
(XLSX)

## Acknowledgments

We thank all the members of the Helaine lab for their comments on the manuscript and fruitful scientific discussions; Melita Gordon and Jay Hinton for providing the clinical isolates used in this study; Helene Andrews-Polymenis, Michael McClelland, and Athanasios Typas for sharing the *Salmonella* Single Gene Deletion library. We also thank Steven Lory for critical reading of the manuscript.

## Author Contributions

**Conceptualization:** Séverin Ronneau, Sophie Helaine.

**Data curation:** Séverin Ronneau, Charlotte Michaux.

**Formal analysis:** Séverin Ronneau.

**Investigation:** Séverin Ronneau, Rachel T. Giorgio.

**Supervision:** Sophie Helaine.

**Writing – original draft:** Séverin Ronneau, Charlotte Michaux.

**Writing – review & editing:** Sophie Helaine.

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
