## [Decision Letter · Decision Letter 0]

21 Nov 2023

Dear Dr Helaine,

Thank you very much for submitting your manuscript "Intoxication of Antibiotic Persisters by Host RNS Inactivates their Efflux Machinery During Infection" for consideration at PLOS Pathogens. As with all papers reviewed by the journal, your manuscript was reviewed by members of the editorial board and by several independent reviewers. In light of the mixed reviews (below this email), we would like to invite the resubmission of a significantly-revised version that takes into account the reviewers' comments.

We cannot make any decision about publication until we have seen the revised manuscript and your response to the reviewers' comments. Your revised manuscript is also likely to be sent to reviewers for further evaluation.

Sincerely,

Jose Luis Balcazar, Ph.D.

Academic Editor

PLOS Pathogens

Nina Salama

Section Editor

PLOS Pathogens

Kasturi Haldar

Editor-in-Chief

PLOS Pathogens

orcid.org/0000-0001-5065-158X

Michael Malim

Editor-in-Chief

PLOS Pathogens

orcid.org/0000-0002-7699-2064

Reviewer's Responses to Questions

**Part I - Summary**

Reviewer #1: In this impressive and detailed study, the authors set out to investigate a puzzling observation – release of reactive nitrogen species (RNS) in macrophages increases production of persisters tolerant of b-lactams in intracellular Salmonella, but these persisters are vulnerable to a fluoroquinolone. RNS inhibit respiration, which causes partial dormancy, inhibits growth, and explain tolerance of b-lactams. The authors now show that the same decrease in energy, specifically pmf, diminishes the activity of the AcrAB/TolC MDR pump, which increases the level of ciprofloxacin in the cell, “intoxicating” persisters. Specific comments follow.

Reviewer #2: Host-derived reactive oxygen and nitrogen species (RNS) help control intracellular bacterial pathogens but can also antagonize antibiotic efficacy through collapsing bacterial TCA cycle and promoting persister cell formation. Enhanced tolerance to multiple different antibiotics under stressful conditions such as the intracellular niche has been demonstrated across a range of bacterial pathogens, including Salmonella. In this study, Ronneau and colleagues observed that intracellular Salmonella was more susceptible to ciprofloxacin than cefotaxime, and its tolerance to ciprofloxacin was not responsive to the RNS level. The authors hypothesize that host-derived RNS facilitates ciprofloxacin killing of intracellular Salmonella by suppressing bacterial respiration and the AcrAB–TolC efflux pump, causing the accumulation of ciprofloxacin in the bacteria. Mutants with defective efflux pump (ΔacrB, ΔtolC) and TCA cycle (ΔsucB) rendered reduced survival rates of intracellular bacteria after ciprofloxacin treatment. Extracellular bacteria exposed to exogenous RNS exhibited reduced efflux rate as reflected by the higher Hoechst fluorescent signal. Lower Hoechst efflux was also observed in the ΔacrB and ΔsucB mutants.

Reviewer #3: The manuscript by Ronneau S. et al. explores the mechanism underlying a recent observation, that reactive nitrogen species (RNS) sensitizes Salmonella to fluoroquinolones. They show RNS targeting of the TCA cycle results in reduction of proton-motive force, preventing ciprofloxacin efflux from the bacterial cell. While ciprofloxacin accumulation was not measured directly, the authors utilized multiple approaches to detect efflux activity and correlate the reduction in efflux activity with ciprofloxacin sensitivity. They also utilized a complemented strain to show altered sensitivity is due to efflux. They incorporated clinical isolates into experiments as well, to show the relevance of these pathways for clinical strains.

The findings here provide an important advance in our understanding of antibiotic tolerance, and combating antibiotic tolerance cells, and the authors systemically explore several different aspects of their model through each Figure. The experiments are well-designed, and strongly support the authors conclusions. I have one suggestion for an additional experiment below with the acrAB complemented strain that I think could strengthen their model further, however the authors have shown the more critical complementation of the survival defect with this strain. This manuscript is extremely well-written, and I only have minor comments below:

**Part II – Major Issues: Key Experiments Required for Acceptance**

Reviewer #1: none

Reviewer #2: Major comments

1. Active efflux pumps have been shown to reduce ciprofloxacin killing in planktonic Salmonella culture. The authors nicely demonstrated that the AcrAB–TolC efflux pump also contributed to bacterial survival under ciprofloxacin inside macrophages. However, with the existing data, it may be difficult to reach the conclusion that host-derived RNS was causing the inactivation of bacterial efflux pump and the enhanced ciprofloxacin killing in macrophages. If the host-derived RNS was the major factor, a higher survival rate should be observed in Nos2 knock-out macrophages compared to the wild-type macrophages, while no difference in the survival rates among Nos2 knock-out and wild-type macrophages with and without IFN-gamma was detected, suggesting that the better killing with ciprofloxacin was RNS-independent. Although the authors demonstrated that exogenous RNS could interfere with the efflux in extracellular bacteria, host-derived RNS may not be the reason for the better efficacy of ciprofloxacin.

2. The differential susceptibility of intracellular Salmonella to ciprofloxacin and cefotaxime could be resulted from their ability to penetrate mammalian cell membranes and accumulate in macrophages. The authors may want to address if the intracellular bacteria were exposed to a sufficient and comparable amount of the two antibiotics in the intracellular niche.

3. It may not be surprising that ΔacrB and ΔsucB mutants extracted from Nos2 knock-out (low RNS) macrophages had suppressed Hoechst efflux as shown in Figure 3F. However, to probe if RNS really impacts the efflux in the intracellular bacteria, the comparison between bacteria extracted from Nos2 knock-out (low RNS) and stimulated (high RNS) macrophages would be needed. The authors could also use efflux pump inhibitors to probe the efflux activity in RNS-high and RNS-low macrophages.

Reviewer #3: • Since the authors have generated a complemented strain (ΔacrB + pacrAB, in Fig. 3G), it would also be important to show this strain no longer has increased accumulation of H33342. It was great to see the authors utilize this strain to complement the survival phenotype, but I think also showing the impact on H33342 accumulation should be included.

**Part III – Minor Issues: Editorial and Data Presentation Modifications**

Reviewer #1: 1. Fig. 1E shows a schematic of a patients infected with S. typhimurium. This creates an impression that the experiment was performed in humans, please drop it.

2. Fig. S1 shows a schematic depicting the experimental approach, please note this in the text.

3. Fig. 2 shows the level of nitrate, please explain how this represents RNS. Nitrate is virtually absent in Nos2- cells, but the difference in wild type cells +/- INF-g used throughout this work is modest, please explain.

4. The authors note that the number of persisters correlates tightly with the level of RNS, but such a correlation is not documented experimentally. One can make a qualitative assumption that RNS correlates with persisters, but “tightly” implies a quantitative measurement. Either provide the measurement or rephrase.

5. Fig. 3D, please add a notation to the y axis to specify what is being measured; currently, it states “fluorescence”, not clear of what.

Reviewer #2: Minor comments

1. In Figure 1E, data of Nos2 knock-out macrophages infected with the clinical isolates after ciprofloxacin treatment should be included for comparison.

2. The inconsistent intracellular survival rates of the acrB mutant under ciprofloxacin in Figure 2C and Figure 3G were noticed.

Reviewer #3: • Fig. 2E: The GFP intensity quantification is described as ‘translational activity’, but really would be a combination of transcription, translation, fluorescent protein folding, and fluorophore maturation. I think a term like ‘Induction capacity’ or ‘Induction ability’ might more accurately reflect GFP intensity here.

• Fig. 4: The model described here was cited briefly once in the text, and could have been discussed in more detail. Some discussion is already present, and filling in additional ‘Fig. 4’ references in the text would reinforce this by drawing the reader to the Figure. The authors should also consider adding a couple of additional sentences to reinforce some of the ideas in Fig. 4B, particularly the heterogeneity of efflux expression that is depicted here.

• Lines 219-220, Fig. 3: The concentrations of H33342 listed in the Results text (0, 0.2, 1, 2) don’t match the concentrations shown in Fig. 3F (0, 0.2, 0.5, 1). H33342 is written as ‘H33345’ in Fig. 3D and 3F.

• Minor: line 262, missing ‘damage’ after DNA?

• Minor: typo line 173, ‘onlybe’

PLOS authors have the option to publish the peer review history of their article (what does this mean?). If published, this will include your full peer review and any attached files.

Reviewer #1: No

Reviewer #2: No

Reviewer #3: No

Figure Files:

Data Requirements:

Please note that, as a condition of publication, PLOS' data policy requires that you make available all data used to draw the conclusions outlined in your manuscript. Data must be deposited in an appropriate repository, included within the body of the manuscript, or uploaded as supporting information. This includes all numerical values that were used to generate graphs, histograms etc.. For an example see here on PLOS Biology: http://www.plosbiology.org/article/info:doi%2F10.1371%2Fjournal.pbio.1001908#s5.
---

## [Decision Letter · Decision Letter 1]

7 Feb 2024

Dear Dr Helaine,

We are pleased to inform you that your manuscript 'Intoxication of Antibiotic Persisters by Host RNS Inactivates their Efflux Machinery During Infection' has been provisionally accepted for publication in PLOS Pathogens.

Best regards,

Jose Luis Balcazar, Ph.D.

Academic Editor

PLOS Pathogens

Nina Salama

Section Editor

PLOS Pathogens

Michael Malim

Editor-in-Chief

PLOS Pathogens

orcid.org/0000-0002-7699-2064

The authors have addressed all reviewers' comments and made substantial improvements to the manuscript. Many thanks!

Reviewer Comments (if any, and for reference):

Reviewer's Responses to Questions

**Part I - Summary**

Reviewer #1: The Authors adequately responded to the critiques. Excellent study.

Reviewer #3: The authors have addressed all my comments and suggestions.

**Part II – Major Issues: Key Experiments Required for Acceptance**

Reviewer #1: The Authors adequately responded to the critiques. Excellent study.

Reviewer #3: (No Response)

**Part III – Minor Issues: Editorial and Data Presentation Modifications**

Reviewer #1: The Authors adequately responded to the critiques. Excellent study.

Reviewer #3: (No Response)

PLOS authors have the option to publish the peer review history of their article (what does this mean?). If published, this will include your full peer review and any attached files.

Reviewer #1: No

Reviewer #3: No

---

## [Editor Report · Acceptance letter]

16 Feb 2024

Dear Dr Helaine,

We are delighted to inform you that your manuscript, "Intoxication of Antibiotic Persisters by Host RNS Inactivates their Efflux Machinery During Infection," has been formally accepted for publication in PLOS Pathogens.

Best regards,

Michael Malim

Editor-in-Chief

PLOS Pathogens

orcid.org/0000-0002-7699-2064